# Driving Signal and Geometry Analysis of a Magnetoelastic Bending Mode Pressductor Type Sensor

**DOI:** 10.3390/s23208393

**Published:** 2023-10-11

**Authors:** Šimon Gans, Ján Molnár, Dobroslav Kováč, Irena Kováčová, Branislav Fecko, Matej Bereš, Patrik Jacko, Jozef Dziak, Tibor Vince

**Affiliations:** Department of Theoretical and Industrial Electrical Engineering, Technical University of Košice, 040 01 Košice, Slovakia; jan.molnar@tuke.sk (J.M.); dobroslav.kovac@tuke.sk (D.K.); irena.kovacova@tuke.sk (I.K.); branislav.fecko@tuke.sk (B.F.); matej.beres@tuke.sk (M.B.); patrik.jacko.2@tuke.sk (P.J.); jozef.dziak@tuke.sk (J.D.); tibor.vince@tuke.sk (T.V.)

**Keywords:** bending, force, frequency, magnetoelasticity, optimization, sensors

## Abstract

The paper deals with a brief overview of magnetoelastic sensors and magnetoelastic sensors used in general for sensing bending forces, either directly or sensing bent structures, and defines the current state of the art. Bulk magnetoelastic force sensors are usually manufactured from transformer sheets or amorphous alloys. In praxis, usually, a compressive force is sensed by bulk magnetoelastic sensors; however, in this paper, the sensor is used for the measurement of bending forces, one reason being that the effect of such forces is easily experimentally tested, whereas compressive forces acting on a single sheet make buckling prevention a challenge. The measurement of the material characteristics that served as inputs into a FEM simulation model of the sensor is presented and described. The used material was considered to be mechanically and magnetically isotropic and magnetically nonlinear, even though the real sheet showed anisotropic behavior to some degree. A sinusoidal magnetizing current waveform was used in the experimental part of this paper, which was created by a current source. The effects of various frequencies, amplitudes, and sensor geometries were tested. The experimental part of this paper studies the sensors’ RMS voltage changes to different loadings that bend the sheet out of its plane. The output voltage was the induced voltage in the secondary coil and was further analyzed to compute the linearity and sensitivity of the sensor at the specific current characteristic. It was found that for the given material, the most favorable operating conditions are obtained with higher frequency signals and higher excitation current amplitudes. The linearity of the sensor can be improved by placing the holes of the windings at different angles than 90° and by placing them further apart along the sheet’s length. The current source was created by a simple op-amp voltage-to-current source controlled by a signal generator, which created a stable waveform. It was found that transformer sheet bending sensors with the dimensions described in this paper are suitable for the measurement of small forces in the range of up to 2 N for the shorter sensors and approximately 0.2 N for the longer sensors.

## 1. Introduction

Magnetoelastic sensors have been extensively studied and experimented with in the last 60 years. One of the most well-known sensors of this type is the Pressductor (abbreviation for pressure inductor) [1], which is based on the principle of changing the permeability of the material due to mechanical stresses, which in turn changes the mutual inductance between two coils. Essentially a transformer, it has a primary and a secondary winding. The windings are placed in such a way that the coupling between them in the unstressed state is zero, resulting in the absence of induced voltage in the secondary coil. The permeability change in the direction of mechanical stress is proportional to the applied load, which changes the flux path in the sheet, and therefore nonzero voltage is induced in the secondary coil. The RMS voltage is usually measured, which is approximately proportional to the applied load, and the characteristics of the input waveform define the sensitivity and linearity of such a sensor. 

This method of force measurement has been found to be reliable and accurate. The properties of usual magnetic materials come with a dispersion of values that tend to not be specified by the manufacturer (the BH curve, the dependence of the magnetostriction coefficient on the applied magnetic field H, etc.); therefore, measuring the characteristics of samples is essential but time-consuming [2]. Moreover, the temperature of the material is affecting its properties, which should be considered [3].

Two distinct sensing approaches are being continuously researched. One approach consists of magnetoelastic thin wire or ribbon sensors, which are, for all practical purposes, geometrically one-dimensional objects that allow the magnetic flux to propagate in only this direction. They are usually manufactured from amorphous ferromagnetic materials by rapid quenching. They are characterized by a large magnetoelastic coupling coefficient [4] and provide a basis for the measurement of various quantities besides mechanical stress such as atmospheric pressure [5], temperature [6], viscosity, chemical composition of the surrounding environment, and more [7]. Their sensing principle is based on the shift of the magnetoelastic resonance frequency due to changes that the environment imposes on the ribbon, thin wire, or special coating that covers their surface. The resonance is also shape-dependent, and variations of the sensor shapes were tested [8]. Their sensing capabilities have been summarized in exhaustive reviews [4,9]. 

The second type of sensors are “bulk” magnetoelastic sensors, which generally can be defined as 3D (or in some cases, 2D) objects in which the magnetic flux can propagate in multiple spatial directions and therefore is not geometrically restricted. The Pressductor sensor belongs to this group. The materials used for their construction are ferromagnetic transformer steels [1], and ferrimagnetic materials [10,11]; however, amorphous materials [12,13] and regular construction steels [14] have also been used. Isotropic hot-rolled transformer steels [1,15] are preferred for their construction because their magnetizing characteristics vary less with the magnetizing direction, which results in zero induced voltage when the primary and secondary coils are placed perpendicular to each other. Anisotropic materials were also used for Pressductor sensor construction, which resulted in a non-zero output in the non-stressed state [16], which was corrected by cutting the sensor in such a way that one diagonal of the square sensor body coincided with the direction of easy magnetization. Other commonly used geometries for such sensors are “ring” and “frame” shapes, mainly constructed for the measurement of compressive and tensile forces [17]. Tensile forces in ferromagnetic sheets were also measured with a non-contact method [18]. The Pressductor shape was analyzed to be used as a bending force sensor, called the “Bendductor” by the respective authors [19] who studied the behavior of a Pressductor sensor being deformed by bending, and several quantities besides the RMS voltage value were measured that were found to be linearly stress-dependent. Cylindrical magnetoelastic sensors were also used for force sensing [20].

When it comes to sensors that are used solely in the bending mode, the current research focuses mainly on the mentioned one-dimensional structures of amorphous thin wires made from Metglas 2826 MB and similar alloys. A beam that is made from aluminum or other construction materials changes shape due to bending, and therefore one side of the beam experiences tensile stresses while the opposite one experiences compression. Magnetoelastic ribbons or wires are anchored to the beam’s surface and therefore are used as tensile or compressive force sensors that relate the tensile or compressive stress to the bending of the beam, which is proportional to the applied force [21]. Such systems are also used for structural health monitoring of cantilever beams because the vibration modes of beams change when cracks and holes start to form in the beam [22,23]. In recent years, the magneto-mechanical ΔE effect has also been used as a sensing mechanism for sensing the bending of structures. Composite materials with a more complex structure are used for the measurement of bending forces, where magnetostrictive materials are placed on a substrate rather than the whole sensor being made from a ferromagnetic material [24]. Sensing vibrations is a typical use of such devices [25]. An indirectly used magnetoelastic effect to measure bending forces is presented in [26].

The only studies to this day that incorporate “bulk” magnetoelastic sensors to measure bending forces are regarding a circumferentially magnetized tube along which surface magnetic poles emerge when stress is applied, and the corresponding magnetic field strength is proportional to the applied stress [27]. Manufactured sensors are made from a sandwich of magnetic and non-magnetic parts, where during bending only the magnetic part is exposed to compression and made from a stress-sensitive material [28]. Pressductor-type sensors have been studied for the use of bending force measurements in which multiple Pressductor sensor windings are placed in the same sheet for increased sensitivity [29]. To this day, the only paper that dealt with the original Pressductor design for the measurement of bending forces is the “Bendductor” analysis presented by Grenda et al. [19] which analyzes the use of the Pressductor sensor for the measurement of bending forces. It was found that various frequencies and waveforms have different effects on the linearity, hysteresis error, and other measurable characteristics of such sensors. The total harmonic distortion (THD) analysis of the voltage waveform was shown to be linearly dependent on the loading of the sensor [19]. However, no papers are currently present that analyze the effect of different coil placement variations on the original Pressductor design and the effect it has on the linearity and sensitivity of the sensor. In this paper, the original Pressductor shape is changed, and we have found that placing the coils at a non-perpendicular angle between them yields a more linear response to loading. Further research in this direction could improve the sensing characteristics of bending mode. Pressductor sensors for them to be used in the industry more commonly. 

For example, to increase the sensitivity of a bending-force Pressductor sensor, one can lengthen the ferromagnetic sheet in one direction to form a prolonged rectangle, making the sensor easier and more prone to deformation while applying the same force, resulting in a configurable sensitivity. Adversely, that means that the yield point will be reached at lower mechanical loadings at which plastic deformation occurs, which irreversibly changes the sensor’s characteristics [30].

An advantage of the Pressductor sensor design is that during the process of cutting the sensor out of a larger ferromagnetic sheet, the cut areas (the sides of the square or rectangle body) will undergo local changes due to stresses that arise during the cutting process in the microstructure, which manifest themselves in the change of magnetic properties. It was specified that the sheet would be damaged due to cutting (or punching) at a depth of approx. 5 mm from the cut area (even though there is no official consensus) [31]. Because the windings are placed inside the sheet far away from the edges, the cutting process does not need to be as carefully executed. However, the holes for the windings should be created with care and then heat treated, since in those places the magnetic flux concentrates and local maxima of mechanical stress are present [16]. In general, waterjet cutting was found to be superior to mechanical cutting, which is in turn superior to laser cutting for the minimization of internal structural changes due to sensor manufacture [32].

Another important advantage of magnetoelastic sensors in general compared to resistive strain gauges, which are more commonly used for force measurements, is that their output signal is several orders of magnitude stronger and that they are superior at handling harsh environmental conditions since the sensing elements of resistive strain gauges are resistive strips, which need to be secured to a deforming body with special glues, whereas magnetoelastic sensors consist of copper coils and a rigid steel body, which are not connected [15]. Further optimization of magnetoelastic sensors could make the differences between magnetoelastic and resistive load cells smaller; therefore, they could operate interchangeably in everyday life. Usually, FEM simulations are used for optimization, which tests the effect of various changes to the geometry and driving parameters from the perspective of linearity and other parameters [33]. Analytical expressions for the sensor’s output voltage are available; however, only for basic shapes such as cylindrical rods, because of the mathematical complexity of the problem [34]. Driving frequency strongly affects the magnetic field distribution of electrical steels due to eddy current and displacement current shielding [35]. A trustworthy parametrized simulation model is essential when it comes to testing many sensor variations without the need to experimentally manufacture and measure them. This approach is also usually avoided because the simulation times can quickly become unreasonable the more complex and nonlinear the model becomes. Great attention should be paid and then verified to make sure that the simplified simulation model captures the behavior of the real physical system. 

In this work, a simulation model of magnetoelastic sensors is created, and some electrical and mechanical quantities needed from simulations are measured with the help of the COMSOL Multiphysics 6.0 FEM software. Magnetic hysteresis is neglected in the simulations. The obtained simulation results are then compared to experimental ones. Extensive experimentation will be described on how the magnetizing current characteristics and winding hole placements affect the linearity and sensitivity of such sensors.

## 2. Materials and Methods

The used materials for the sensor construction were silicon steel sheets in the form of I and E cut-outs with a thickness of 0.3 mm, which were readily available at the Department of Theoretical and Industrial Electrical Engineering (DTIEE) at the Technical University of Košice (TUKE). The material name is unknown, with no relevant data available about the manufacturer; therefore, all relevant characteristics that describe it needed to be determined experimentally. It was known that structurally, it is an electrical steel because it is rather brittle and cracks easily when excessively bent compared to regular steel. This is typical for silicon steels, which usually have an approximate silicon content of 4%. Materials for transformer cores are mostly anisotropic and are cut in a way that would benefit from the easy-magnetization direction, which is the rolling direction [36]; therefore, a degree of anisotropy is to be expected. The material characteristics will be presented in successive chapters. 

For the winding of all coils, enameled copper wire with a diameter of 0.3 mm was used. It is important to mention that before the winding process, padding was inserted on the edges of the cut holes in the material so the wire’s insulation would not get damaged during the winding process, causing shorts between neighboring coil turns. Standard electrical tape was used for this purpose. All coils (but especially the larger coils) were secured in place to the sensor with epoxy so movement would be prohibited. The shift of coil turn positions could affect the induced voltage characteristics of the sensor and therefore change the sensor entirely. 

The simulation software used was COMSOL Multiphysics version 6.0 with the magnetostriction Multiphysics module, which was used to create a bidirectional coupling between magnetic fields and solid mechanics simulation. The model of the sensor was created in the Fusion 360 3D modeling software (version 2.0.16761) and then imported into COMSOL Multiphysics via a sat file, and then the respective 3D domains were assigned their material characteristics. 

For the magnetizing primary coil, a voltage-controlled current source was created, which will be presented in Section 5.

## 3. Material Characteristics Determination

In this chapter, the estimation process for the material characteristics will be presented. One material characteristic that will be guessed without being measured is the Poisson’s ratio *µ* of the transformer steel. Transformer steels were measured to have a *µ* value of approximately 0.3 [37,38]. The same value will be used in this work too.

### 3.1. Estimation of the Young’s Modulus

To fully specify the elastic mechanical behavior of the material, in addition to the Poisson’s ratio, the Young’s modulus *E* must also be specified. The steel was mechanically isotropic. The estimation method was inspired by [39]. A 7-cm-long transformer steel “I” cutout was chosen and modified (Figure 1). Because of the introduced holes, which affect the stress distribution [40], it needed to be modeled via FEM software to predict the beam deflection when loaded. The Fusion 360 modeling software was used to model the I cut out. 

A string was placed through the notches, and a weight of 127 g was connected to the string. Assuming that *g*, the gravitational constant, is approximately 9.81 m·s^−2^ the total bending force was 1.246 N. The simulation model is shown below (Figure 2), and the output of the simulation was the z-coordinate change of the free (yellow) end. The experimental value was obtained by measuring the difference in the height over the ground of the free end of the sheet in the loaded and unloaded states (Figure 3).

A deflection of approximately 14 mm was observed. The displacement was measured with a measuring tape with a resolution of 1 mm; therefore, if no parallax error occurred, the absolute accuracy of the measurement is ±1 mm. 

Because Young’s modulus and Poisson’s ratio define the elastic mechanical behavior of the sheet fully and Poisson’s ratio has been chosen beforehand, changing Young’s modulus values in the simulation software to fit the observed deflection of the free end (Figure 4) should yield the real value of *E*. For the observed displacement value of 13 ± 1 mm, the *E* value ranged from 185.819 GPa to 218.151 GPa. The default (Nelder-Mead) optimization method in the COMSOL Multiphysics optimization module was used to match the simulation to the experiment. The measured values for electrical steels from other researchers fall into this interval [41], and a middle value of 202 GPa was taken for simulations. The experiment is described in further detail in [42]. 

### 3.2. Estimation of the Electrical Resistivity

The electrical DC resistance of the sheet was determined by using the 4-wire method. The experiment is described in [43] in more detail. A chosen sheet was modified by soldering copper wires to the sheet, acting as terminals (Figure 5). A resistance of approximately 10.5784 mΩ was measured at an electrical current flow of 1 A. 

A FEM model of the sheet was created, to which the physics of electrical current flow was added (Figure 6). The resistivity of the material was set as isotropic and varied until the experimental voltage drop was obtained. The methodology was inspired by [44].

An electrical conductivity of approximately 2.246 MS/m was estimated with an estimated error of 3.5%. For simulations, a conductivity of 2.25 MS/m was used. This value is consistent with the conductivities defined in the COMSOL Multiphysics nonlinear magnetic materials library for oriented and non-oriented silicon steels, which range from approx. 1.7 MS/m to 4 MS/m [45].

### 3.3. Estimation of the Anhysteretic Magnetization

The specific material type was not known, and therefore the B-H curve had to be measured as well. The most probable supplier of the transformer sheets (located in Frýdek Místek in the Czech Republic) no longer exists, and therefore datasheets of materials cannot be obtained. The B-H curve was measured at a magnetizing current frequency of 0.1 Hz, at which the material behaves quasi-statically. The experimental setup is shown in the picture below (Figure 7). 

Two E-shaped cutouts were placed against each other to form a closed magnetic circuit. Coil bobbins were designed in the Fusion 360 3D modeling software for the magnetizing and sensing coils and were printed on an Ender 3 Pro 3D printer (Figure 8). A standard PLA material has been used. The magnetizing coil had 567 turns of enameled copper wire with a diameter of 0.5 mm, and the sensing coils were wound with 156 turns of wire with a diameter of 0.3 mm. 

The measurement of open-ended samples can be carried out as well, but compensation for the stray fields must be carried out by employing numerical methods whose accuracy depends on the shape and dimensions of the open sample; therefore, closed magnetic circuits are preferred. The magnetizing current was provided by a current source that consisted of a dedicated audio amplifier IC that was controlled by an HMF 2550 signal generator, creating a sinusoidal voltage waveform with a frequency of 0.1 Hz. The primary current amplitude was 1 A. The primary current was measured by a precision 10 Ω shunt resistor. The sensing coil voltage was connected to a Lakeshore 480 fluxmeter, which integrated the waveform with respect to time and yielded a waveform directly proportional to the magnetic flux flowing through the material. The waveforms can be seen below (Figure 9). Due to the non-harmonic square wave-like fluxmeter output, it is evident that the current magnetized the material into magnetic saturation; therefore, the major BH curve was measured. 

For the magnetic field simulation model in COMSOL Multiphysics, the anhysteretic magnetization function must be specified. The curve is directly experimentally attainable, but it is carried out through a rather time-consuming and experimentally challenging process [46]. For technical applications, the anhysteretic curve can be approximated by interpolating the “middle” curve of the hysteresis loop [47]. The graph of the measured BH curve and the anhysteretic magnetization estimation are shown below (Figure 10). The measured curve was also filtered by using a moving mean function with a window size of 20 (Figure 11). Two periods of the 0.1 Hz signal were obtained, yielding 100,000 data points.

For an analytic expression of anhysteretic magnetization, the Langevin function is commonly used, which describes the anhysteretic magnetization process of soft magnetic materials well. The equation can be seen below (1),
(1)$$M_{an}\left(H\right)=M_{S}\left(\coth\left(\frac{3\chi_{0}H}{M_{S}}\right)-\frac{M_{S}}{3\chi_{0}H}\right)$$
where *M_an_* is the anhysteretic magnetization, which is a function of the magnetic field strength. *H*, *χ*_0_ is the initial magnetic susceptibility, which is the slope of the magnetization curve near the origin, and *M_S_* is the magnetization obtained at material saturation [48], which can be estimated directly by using the well-known relationship (2),
(2)$$B=\mu_{0}\left(H+M\right)$$
and from which the saturation magnetization can be obtained by Equation (3) by taking a data point that is well within the saturated region of the B-H curve, ideally its tip,
(3)$$M_{S}=\frac{B_{S}}{\mu_{0}}-H_{S}$$
therefore, in Equation (3), *B_S_* and *H_S_* are the coordinates of the tip of the magnetization curve, and *µ*_0_ is the permeability of the vacuum. An *M_S_* value of approximately 1.4 MA/m was calculated. The initial susceptibility *χ_0_* can be estimated by fitting the theoretical expression (1) to the interpolated middle curve. The nonlinear least squares method in the MATLAB 2021b software was used (*lsqnonlin*) for this task. An initial susceptibility value of 1419.2 was estimated. The values *M_S_* and *χ*_0_ fully specify the anhysteretic magnetization curve, which is shown below compared to the filtered BH curve (Figure 11).

The magnetostriction value was taken from literature for similar materials (silicon transformer steels) and was given a value of 20 ppm. The magnetostriction value in the COMSOL software directly specifies the sensitivity of the material to stress; therefore, the real value is approximated better by fitting the measured sensor waveforms to the simulations. Using this value, a good agreement between simulation and experiment was obtained, which is shown in Section 6.

## 4. Sensor Samples

The main purpose of this paper is to analyze the effect of hole placement on the linearity and sensitivity of the Pressductor sensor to bending forces. Multiple sensor samples were manufactured from the transformer sheets with differently placed holes. The Pressductor type sensor principle is explained by the pictures below (Figure 12), and further information can be found in the original paper [1]. 

If the sensor is given isotropic linear properties, the largest sensitivity (voltage change per Newton of force) is obtained when the primary and secondary coils are placed perpendicular to each other. When anisotropy is introduced, the no-load, zero-voltage condition no longer holds for this configuration since the permeability in two perpendicular directions is not the same. For example, in Figure 13, if the permeability in the *y* direction *μ_y_* was set to 600 and in the *x* direction *μ_y_* to 400, then the windings needed to be shifted by 1.2 mm further from each other in the y-direction to produce the zero output at no load (the original Pressductor shape has holes placed in the vertices of an imaginary square).

In the paper by Nowicki [16] it was shown that when manufacturing Pressductor sensors from anisotropic sheets, the orientation of the sensor during cutting relative to the easy magnetization direction can be adjusted to maintain the zero voltage, no load characteristic. This holds if symmetric anisotropy is present around the easy magnetization direction. 

In our work, the effect of changing the hole placement on linearity and sensitivity is investigated. The transformer sheets were manufactured into long, narrow strips. Multiple sensors were cut from the strips, which can be seen below (Figure 14). The sensors will be referred to as (S-a), (S-b), and (S-c) from now on. The dimensions of the sensors were 31 mm × 16 mm, and the diameters of the holes were 3 mm. The holes of the (S-a) sensor were placed in a 3 mm × 3 mm grid. Each sensor was subjected to different external loadings at various magnetizing current amplitudes and frequencies. The primary coils were created by 20 turns of copper wire with a diameter of 0.3 mm, and the secondary coils were wound with 15 turns of the same wire. 

Alongside them, longer sensors were cut (48 mm × 16 mm), which should be more sensitive to loads because they exhibit greater deformation at the same load due to a more bendable geometry. Two samples were created, which will be from now on referred to as (L-a) and (L-b) (Figure 15).

The samples were tested under incremental loading, and the linearity of their loading characteristics and their sensitivities have been computed, which will be presented in Section 6.

## 5. Experimental Setup

The experimental setup for the sensor characterization of RMS voltage changes due to mechanical loading that bends the sensors can be seen below (Figure 16). 

A voltage-controlled current source has been created to supply the magnetizing current to the primary coil. The current source was soldered to a perf-board (Figure 17). Two complementary BJT transistors (TIP 41 C and TIP 42 C) are used as current regulating elements. The voltage drop across a 0.1 Ω shunt resistor is amplified by a non-inverting LM358 operational amplifier. The amplified signal is compared with the waveform from an HMF 2525 arbitrary signal generator via an LF3055 operation amplifier used as a comparator. Its output controls the bases of the BJT transistors via a 50 Ω resistor. The secondary coil voltage was amplified by an LM324 operational amplifier used as an instrumentation amplifier. A combination of 100 nF ceramic and 220 μF electrolytic capacitors are used as buffer capacitors for the op-amps. A cooling fan was added to cool the heatsinks of the transistors during operation.

The amplified secondary voltage and the primary current were measured by an oscilloscope. The RMS voltage was automatically measured on the Rigol DM3058 multimeter, which stored the measured values on a USB stick. The loading mechanism of the sensors is shown below (Figure 18).

## 6. Experimental Results

The RMS voltage changes of the (S-a) sensor (the original Pressductor) at a current amplitude of 180 mA and various frequencies are shown below (Figure 19). A nonzero voltage is induced in the sensing coil at no load, which shows the anisotropic nature of the material. Multiple 47 g weights were added to the sensor in ascending order at sinusoidal magnetizing current frequencies from 250 Hz to 3500 Hz. Figure 20 shows the RMS voltage changes relative to the unstressed state. As the frequency increases, the sensitivity increases as well, creating larger voltage changes at the same load. 

The graphs from Figure 20 were analyzed further, where a linear regression of the datapoint coefficient of the linear term was calculated, representing the sensor’s sensitivity. The linearity error of the loading characteristic was determined by computing the maximum linearity error [49], which is computed by Formula (4),
(4)$$\delta m=\max\left(\left|\frac{am-\Delta U_{RMS}+b}{am_{max}}\right|\cdot 100\right)\;\left[\%\right]$$
where δ*m* is the linearity error in % of the static loading characteristic, *m_max_* is the largest load applied during the measurement, *m* is the applied load of a given datapoint, Δ*U_RMS_* is the corresponding RMS voltage change at the given load, and coefficients *a* and *b* are the linear and constant terms of the linear regression analysis. The analysis yielded the data shown below (Figure 21).

The linearity error of the characteristics grows linearly with increasing frequency, and the sensitivity increases; however, the sensitivity gain diminishes at higher frequencies. A compromise between linearity and sensitivity should be found so that the sensor is sensitive enough while still being linear. One way to characterize this problem is to combine these statistics into one by using Equation (5),
(5)$$y_{c}=ay_{l}+by_{s}$$
where *y_l_* is the normalized linearity characteristic, *y_S_* is the normalized sensitivity characteristic, *a* and *b* are the weights given to those characteristics, and *y_C_* is the combined characteristic of the sensors. The characteristics are normalized by using (6),
(6)$$y_{norm}=\frac{y-\min\left(y_{1},y_{2},\ldots ,y_{n-1},y_{n}\right)}{\max\left(y_{1},y_{2},\ldots ,y_{n-1},y_{n}\right)-\min\left(y_{1},y_{2},\ldots ,y_{n-1},y_{n}\right)}$$
where *y*_1_ to *y_n_* are the performance metrics in the set of data points. Each characteristic from the tested set is given a number between 0 and 1, linearly spaced based on performance in the given set. The coefficients *a* and *b* were chosen to be −1 and 1 since a lower linearity error and a higher sensitivity are more desirable. After applying this analysis to the measurements, the graph below is created (Figure 22).

Naturally, the graph of the combined characteristic and therefore the most favorable frequency is dependent on the chosen coefficients *a* and *b*, which depend on the use case. 

### 6.1. The Effect of Different Placement of Holes on the Static Response (S—Sensors)

The sensors (S-a), (S-b), and (S-c) were tested to analyze the effect of hole placement on the loading characteristics. Two magnetizing current amplitudes—300 mA (Figure 23 and Figure 24) and 100 mA (Figure 25 and Figure 26)—were used. 

When analyzing the linearity and sensitivity of the three sensors, as shown in Section 5, the following graphs were created (Figure 24 and Figure 26).

The lower-amplitude 100 mA responses showed a nonlinear initial region that was not present at the 300 mA excitation. This could be solved by placing a 100 g preload on the loading platform.

### 6.2. The Effect of Different Placements of Holes on the Static Response (L—Sensors)

When sensor (L-a) was tested with different combinations of magnetizing current amplitudes and frequencies, which were further analyzed, the following graphs were created (Figure 27 and Figure 28). An increase in linearity with an increase in current amplitude was observed. In all cases, it seemed like with increasing amplitude, the initial linear region was extended (Figure 27). Their linearity and sensitivity analyses yielded the data shown in Figure 28.

At higher current amplitudes, sensitivity and linearity increased. This trend will be maintained until the amplitude does not start to significantly saturate the material. When a ferromagnetic material is saturated, its permeability reduces drastically, effectively having a permeability close to vacuum. The change in permeability due to stress is negligible at this point. 

Supplying higher current amplitudes can put a strain on the internal components of the power source, and high-power transistors with sufficient specifications, such as the nominal drain/collector current, should be used, which increases costs. Using a magnetizing winding that has more coil turns creates stronger magnetic fields at the same current, and the coil can be designed using FEM simulation software to obtain the magnetic field strength at a lower current. More coil turns result in higher losses in the DC resistance of the coil, which can therefore reach higher temperatures during operation. The rising temperature can change the permeability of the sheet since the magnetic properties of ferromagnetic materials are strongly temperature-dependent. The rising temperature of the coil will increase its resistance, and because the same current will be maintained by the current source, the power loss will rise further. The power loss can be optimized by minimizing the resistive losses characterized by Equation (7),
(7)$$P=R\cdot I^{2}\;$$
where *P* is the power loss, *R* is the resistance of the coil, and *I* is the effective value of the flowing current. Increasing the current twice at the same number of coil turns increases the power loss four times, whereas simply doubling the number of turns increases the power loss only two times. Using thicker magnetizing coils for lower DC resistance is usually not an option since the holes should be kept as small as possible. Larger holes create larger variations in the stress and magnetic field distribution and worsen the mechanical behavior of the sheet, making it easier to overload. For further sensor optimization, it is more desirable to increase the number of turns than to increase the current.

Increasing the current amplitude of the (L-a) sensor further at a frequency of 50 Hz and current amplitudes of 100 mA to 500 mA yielded the graphs below (Figure 29 and Figure 30).

Because the interval of 0–14 g was shown to behave linearly at all amplitudes of the 50 Hz current, the analysis was redone considering only this interval. The following results were obtained (Figure 31).

Because strong nonlinearities were also present in the characteristics from Figure 27, the same truncated analysis (with loads from 0 g to 14 g) was redone for the L-a) sensor. The analysis yielded the following graphs (Figure 32). The linearity errors in this region were not larger than 16.3%. When the combined characteristic analysis *y_c_* was carried out on these graphs, the most favorable frequency excitations were the higher ones, since higher sensitivities and moderate linearity errors were observed when using them, which can be seen in Figure 33.

### 6.3. The Effect of Different Placement of Holes on the (L-b) Sheet

When the (L-b) sensor was measured with a sinusoidal magnetizing current with an amplitude of 300 mA and various frequencies from 50 Hz to 400 Hz, the following RMS voltage change data (Figure 34) and further analysis were obtained (Figure 35).

Interestingly, placing the windings further apart (sensor (L-b)) made the sensor characteristics less dependent on magnetizing current frequency. When we used only the truncated analysis on the sensor characteristics from 0 to 14 g, the following data were observed (Figure 36). The linearity error in that specific region was no larger than 3.24%, and a sensitivity of 31.59 mV/g was obtained. The (L-a) sensor at an excitation amplitude of 300 mA was measured to have a linearity error of approx. 10%. 

## 7. Simulation Results

A simulation model for the sensor labeled (S-a) was created, and several loading responses were computed via the COMSOL Multiphysics software. It was found that solving a 3D system of coupled nonlinear equations between the mechanical field and the electromagnetic field is computationally very expensive. Some special cases allow the system to be simplified to lower dimensions by utilizing symmetries in the system, but in this case, no such simplification could be employed. What was not yet mentioned in published research and analyzed in the available literature is that the bending creates a change in the topology of the magnetic circuit, which in turn changes the magnetic field distribution around and inside the sensor. Physical measurements of the sensor current and induced voltage yielded the graph below (Figure 37). The measurements of the secondary voltages at different mechanical loadings are shown in Figure 38. Voltage waveform differences due to loadings are shown below (Figure 39). 

When such systems (Figure 40) were simulated, the combination of the thin sheet and the force acting perpendicular to the plane of the sheet in combination with the placement of the windings required the system to be described as a 3D problem. A fully coupled formulation of the mechanical field and magnetic field simulation was used (magnetostriction and magnetoelasticity). The high aspect ratio of the sheet thickness and its length-to-thickness ratio required a fine mesh throughout its thickness and the nonlinear geometry model.

Five layers of the surface mesh were swept through the thickness of the sheet (Figure 41), which yielded the results shown below (Figure 42). The bent geometry and the shape of the holes required a quadratic geometric discretization. The computational problem took a single frequency and amplitude and four different loadings approximately 67 h to finish. Still, the mesh seemed to be not fine enough because, with increasing mesh densities, slightly different results were obtained. When facing an optimization problem that usually requires many parameter combinations to be simulated, this approach, with the computational power currently at hand (an Intel i9-9900K processor with 32 GB of RAM) and three other computers (an Intel i5-10300H with 16 GB of RAM), is deemed not sufficient for the task. Simulating a characteristic obtained experimentally like in the figure above (Figure 27) with a mesh of the density that we deemed to be sufficient would take approximately 1 year on the Intel i9-9900k processor to finish, which would describe a response of a single sensor geometry to a single current amplitude value at 15 different frequencies and 20 different loadings. 

An excellent match between experimental data and simulations has been observed for a perfect square sensor, which is depicted in the picture below (Figure 43) and has dimensions of 31 mm × 31 mm. The winding holes were placed in the vertices of a square with a side length of 7 mm. The results are shown in the graphs Figure 44 and Figure 45.

The simulations show promising results, confirming the correctness of the estimated magnetic and mechanical characteristics of the material; however, due to their computational cost, they cannot be used for sensor optimization in the current state. Great simplifications of the model, such as reformulating the ferromagnetic material behavior to a linearized one where the saturation and initial Rayleigh region nonlinearity are neglected, can make simulation times feasible for geometry optimization; however, since the induced voltage waveforms are highly non-linear, it has yet to be tested if there would be a correlation between the RMS voltage changes in the real-world non-linear case and the linearized simulations.

## 8. Conclusions

It was observed that placing the holes at different locations compared to the typical “x” shape in which the windings meet at a 90° angle increased the linearity of the transfer characteristics in all of the tested samples. The experiments conducted on the shorter versions of the sensors (S-a) (original Pressductor design), (S-b) (6 mm offset of the coils), and (S-c) (12 mm offset of the coils) have yielded transfer characteristics whose sensitivity was higher at higher frequencies in all cases. The linearity of the characteristics was decreasing with increasing frequency in the case of the (S-a) sensor, which was also observed in the original Pressductor paper 1 [1], but increasing in the case of the (S-c) sensor, which also yielded a 2.1% linearity error at a current amplitude of 300 mA and a frequency of 250 Hz, which was the best of the tested sensors in the range from 0 g to 200 g. Lower current amplitudes, such as the tested 100 mA, created more nonlinear transfer functions, whose linearity decreased by approx. 6% in the case of the (S-a) sensor, 8% for the (S-b) sensor, and more than 10% for the (S-c) sensor compared to the tests conducted with current with an amplitude of 300 mA. However, even at lower amplitudes, linear regions emerged in subintervals of the given characteristics, especially in the case of the (S-c) sensor meaning that a linear sensor response can be reached at lower current amplitudes at a sufficient preload.

When the longer sensors (L-a) (10 mm offset of the coils) and (L-b) (20 mm offset of the coils) were tested, it was found that for the (L-a) sensor, its sensitivity increased with increasing frequency up to a frequency value of 150 Hz, after which a plateau was reached where the sensitivity was kept nearly constant at approx. 4 mV/g until a frequency of 350 Hz was reached, after which another increase was observed. The linearity error decreased with increasing frequency between 25 Hz and 175 Hz, at which a local minimum was reached for all frequency values with a linearity error value of under 5%. 

As the magnetizing current amplitudes were increased, the output signal of the sensors became more linear in all cases. The introduced combined characteristic has shown that from the linearity and sensitivity viewpoint, considering the (L-a) sensor, the higher frequency input current signals (more than 350 Hz) were optimal since they offered the highest sensitivity values of approx. 5 mV/g and relatively modest linearity errors of approx. 6% at an excitation amplitude of 300 mA. When the sensor (L-a) was tested at a frequency of 50 Hz and different current amplitudes from the range of 100 mA to 500 mA were tested, the higher amplitudes created a characteristic that was defined by a larger sensitivity and lower linearity. At an amplitude of 500 mA, an 8.30 mV/g sensitivity value and a 9.29% linearity error value were measured at loads ranging from 0 g to 40 g. When only the most linear part of the load characteristic (loads from 0 g to 14 g) was taken, the 500 mA amplitude created a sensitivity of 11.65 mV/g and a linearity error of just 2.63%.

The (L-b) sensor showed interesting behavior because of its consistency regarding its linearity, where the linearity error was approximately constant at 9% for all current frequency values and the sensitivity of the sensor monotonically increased with increasing frequency, making the most optimal driving frequency the largest tested one of 400 Hz (at a current amplitude of 300 mA), where the sensitivity was 23.52 mV/g measured in a load range from 0 g to 40 g. When the load range of 0 g to 14 g was analyzed when considering the (L-b) sensor, the sensitivity was linearly increasing with frequency at a current amplitude of 300 mA, and the maximum linearity error observed was at a frequency of 125 Hz of 3.24% and the lowest linearity error of 0.67% at a frequency of 350 Hz. At 400 Hz, the sensitivity obtained was 31.59 mV/g and therefore has more favorable metrological characteristics compared to the (L-a) sensor, which means that putting a larger distance between the holes for the windings so they do not meet at a right angle has a beneficial effect on the linearity and sensitivity of a sensor, which is a result that has not been publicized yet and can open new grounds for sensor optimization besides analyzing the effect of driving frequency and amplitude.

The simulation model creates a good match between the experimental results and the simulation output, but it takes a lot of time, even with moderate computational power, to finish. Continuous work on simulation model optimization, such as finding the optimal mesh density, is being carried out, and trying to work with linearized material characteristics for the simulation is being tested with not very satisfactory results at this time. For now, with the hardware options that we have currently available, manufacturing the sensors from sheets, measuring their responses at various magnetizing current parameters, and analyzing the data are less time-consuming than simulating the sensors virtually when it comes to considerable bending of thin sheets. FEM simulations conducted by other researchers when considering problems similar in nature (magnetoelastic torque sensors, rather than magnetoelastic bending mode force sensors) were conducted, and outputs were defined by a sensitivity chart that showed local maxima of sensitivity in a given range of torque moments but not the actual waveforms [33], which are characteristics that are probably obtainable even from linearized simulation models that do not need to incorporate the truly nonlinear behavior of the material.

One important aspect of bending moment sensors that needs to be considered is that the experimenter must choose sensor loadings carefully beforehand so that the sensor will not be overloaded. Plastic deformation could occur and therefore change the magnetic behavior of the sensor. It can be easily conducted with simple mechanical simulations that analyze the stress distributions inside the sensor and compare them to the elastic limit mechanical stresses for the given material.

## Figures and Tables

**Figure 1 sensors-23-08393-f001:**
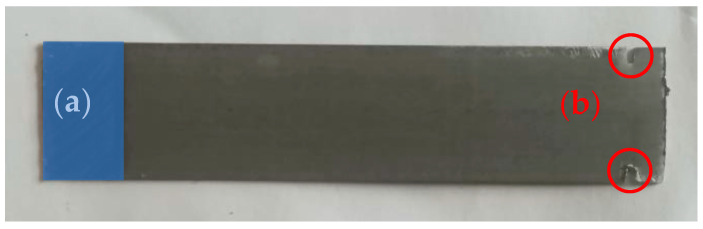
The modified transformer sheet that was used for the estimation of the Young’s modulus. The area denoted (**a**) was clamped to an immovable object and therefore prevented displacement. The area had dimensions of 8 mm × 16 mm. The two notches were marked with red circles (**b**), were placed 3 mm from the end of the sheet, and were 1 mm deep.

**Figure 2 sensors-23-08393-f002:**
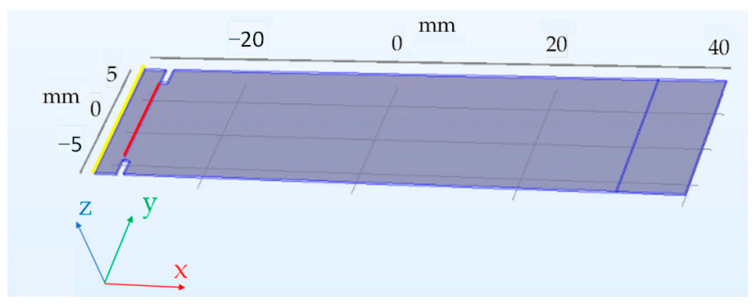
The 3D simulation model of the sheet. The load is assumed to be uniformly distributed along the red line, and the z-axis position of the yellow line is the simulation output.

**Figure 3 sensors-23-08393-f003:**
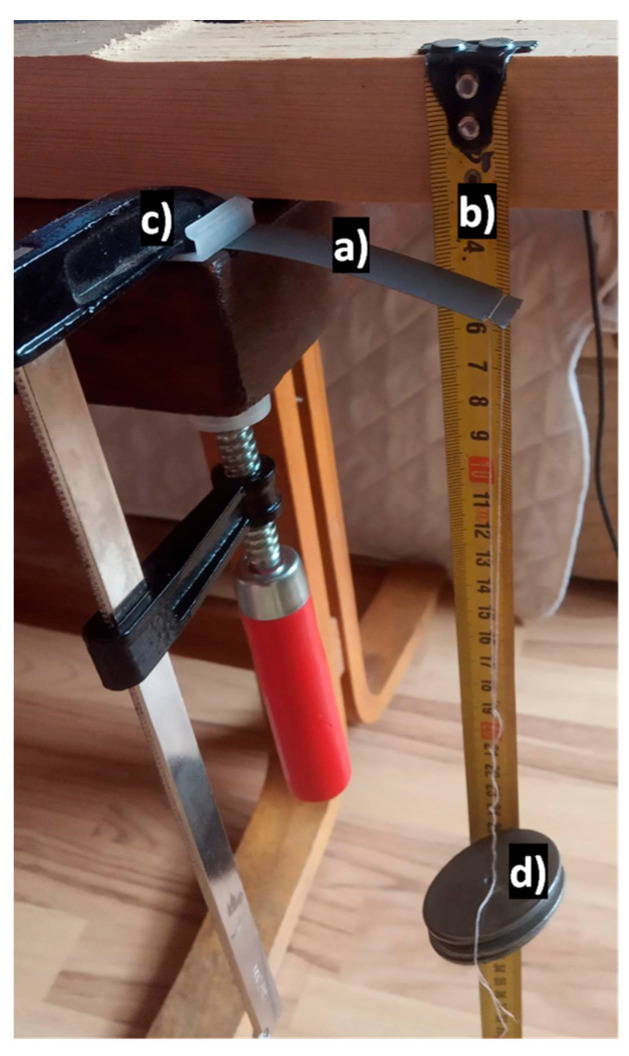
The measurement system; (a)—the transformer sheet; (b)—measurement tape; (c)—clamp; (d)—weight.

**Figure 4 sensors-23-08393-f004:**
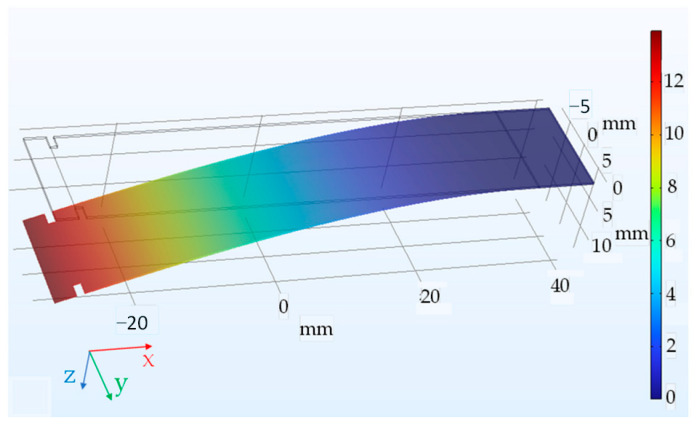
The computed deflection of the sheet under load on the z-axis is in mm when a value of *E* = 202 GPa is set for the Young’s modulus.

**Figure 5 sensors-23-08393-f005:**
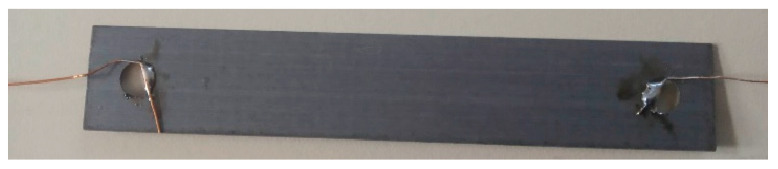
Terminals (copper wire) have been soldered to opposite sides of the sheet. The voltage drop was measured between these two solder joints at a given current flow.

**Figure 6 sensors-23-08393-f006:**
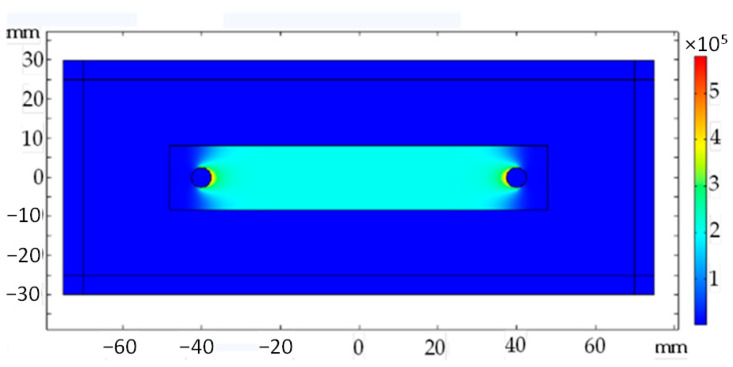
The current density norm is throughout the sheet at a current flow of 1 A. The current norm values are expressed in A/mm^2^.

**Figure 7 sensors-23-08393-f007:**
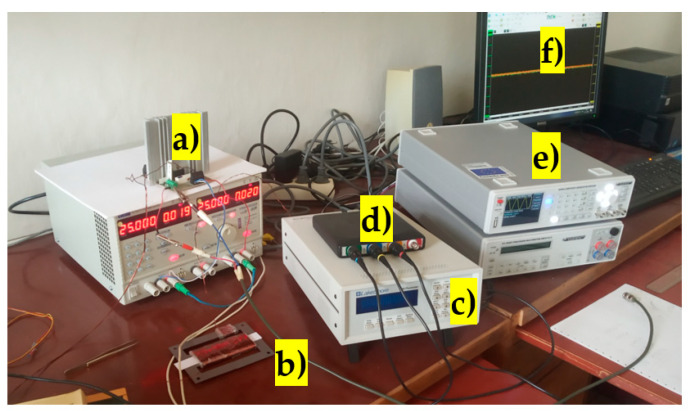
The setup used for measuring the BH curve of the sheets (a) high-current audio amplifier; (b) measured sheet sample; (c) Lakeshore 480 fluxmeter; (d) digital USB oscilloscope; (e) HMF 2550 function generator; (f) PC with the oscilloscope data visualization and analysis software.

**Figure 8 sensors-23-08393-f008:**
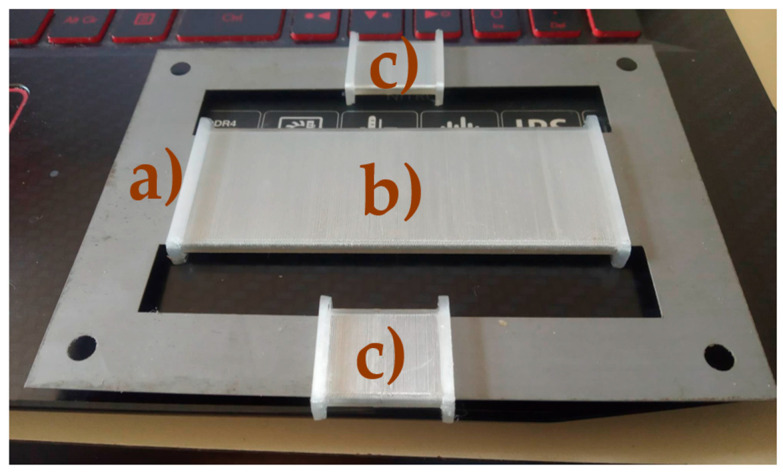
Two E cut-out (a) sheets were placed against each other with bobbins for the magnetizing coil (b) and sensing coils (c).

**Figure 9 sensors-23-08393-f009:**
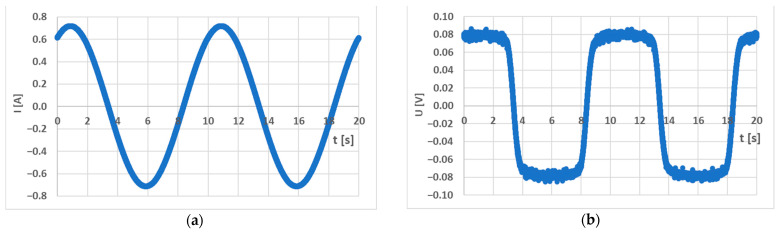
(**a**) The waveform of the current flowing through the primary coil; (**b**) The output voltage signal of the fluxmeter, which is proportional to the magnetic flux.

**Figure 10 sensors-23-08393-f010:**
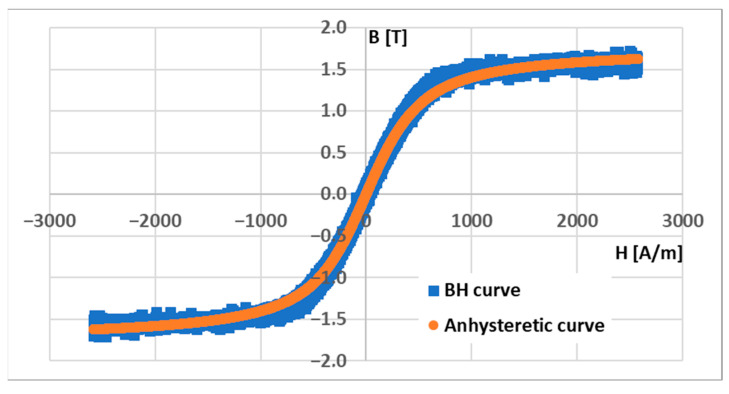
The unfiltered hysteresis loop (blue graph). The Langevin function was used for the interpolation of the anhysteretic curve (orange graph).

**Figure 11 sensors-23-08393-f011:**
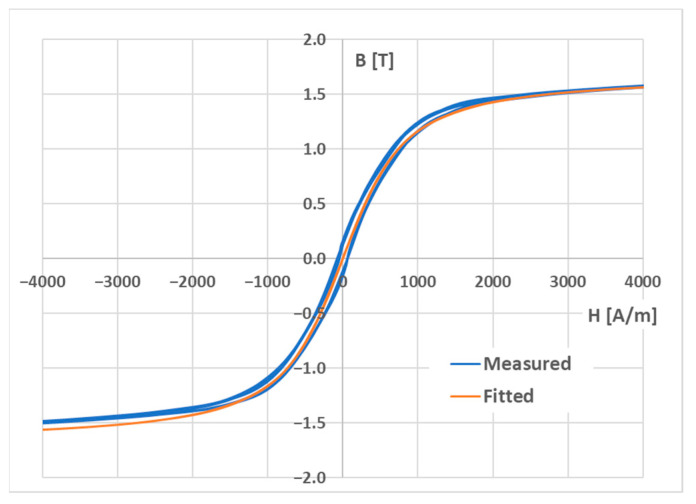
The smoothed BH curve from the measurement (blue) and the estimated anhysteretic magnetization curve (orange).

**Figure 12 sensors-23-08393-f012:**
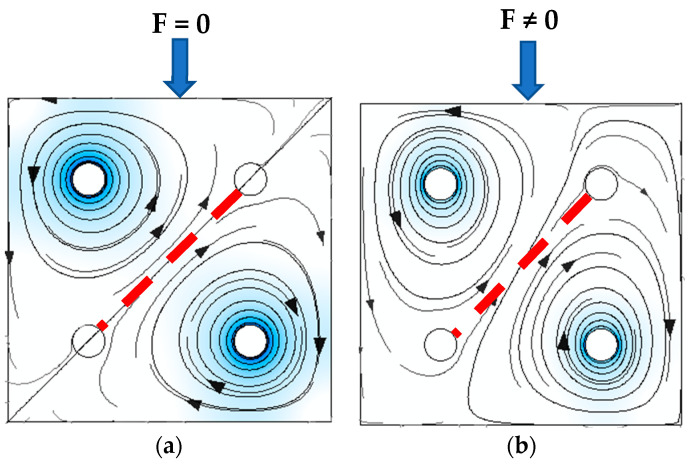
Pressductor principle of operation. The red dashed line represents the sensing coil of the sensor. (**a**) When not stressed, magnetic flux does not flow through the sensing coil, and 0 voltage is induced; (**b**) Stress changes the flux path, directing it through the sensing coil, and voltage is linearly proportional to the acting force.

**Figure 13 sensors-23-08393-f013:**
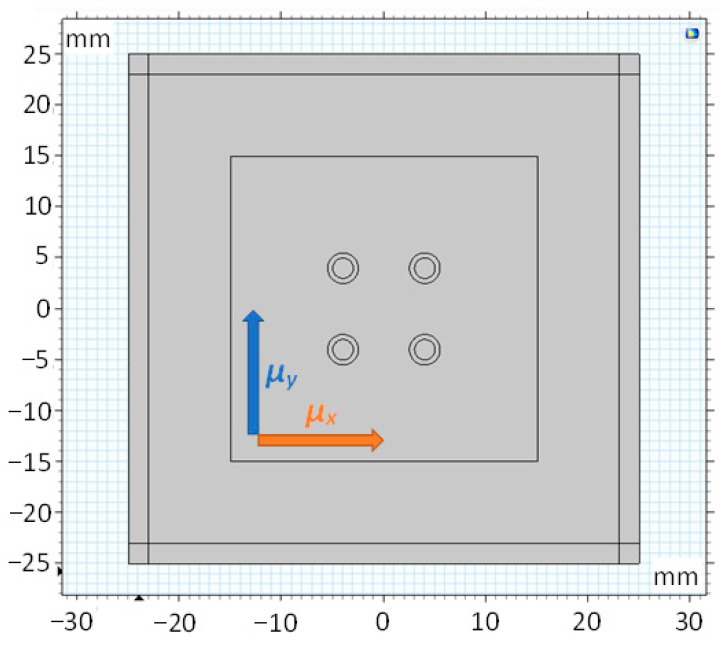
A selected coordinate system for the permeability anisotropy.

**Figure 14 sensors-23-08393-f014:**
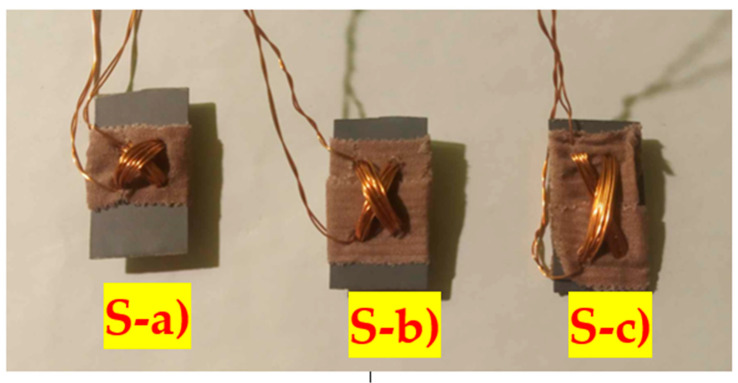
Three samples of the Pressductor sensor each have differently positioned primary and secondary coils: (S-a) perpendicular, (S-b) 6 mm offset in the y-axis, and (S-**c**) 12 mm offset in the y-axis.

**Figure 15 sensors-23-08393-f015:**
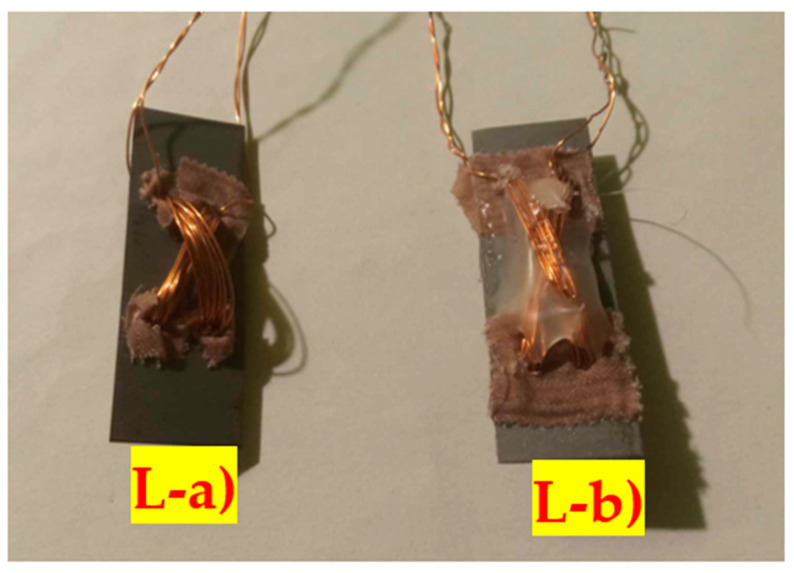
Two longer versions of the Pressductor bending mode sensors are (L-a) 10 mm offset in the y direction and (L-b)—20 mm offset in the y direction. The sample (b) had its windings fixed using glue, which reduced the drift of the sensor due to winding movements.

**Figure 16 sensors-23-08393-f016:**
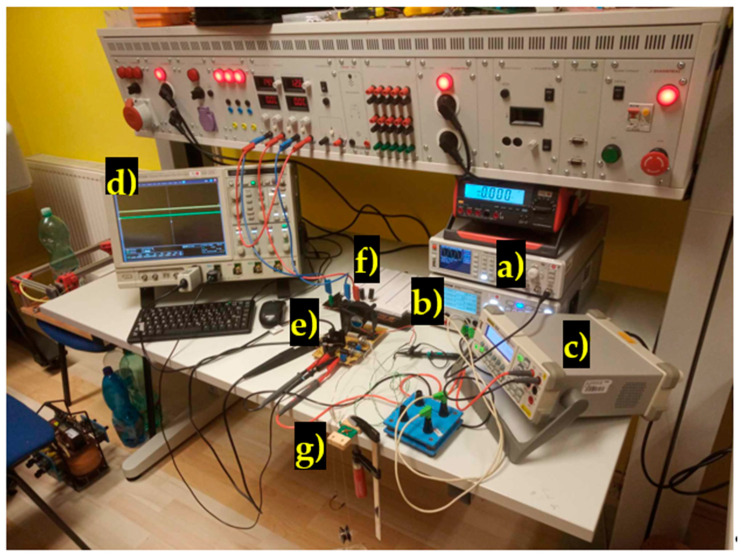
The experimental measurement setup. (a) Signal generator; (b) 4-channel DC voltage source; (c) Rigol DM 3068 RMS voltmeter; (d) Tektronix Oscilloscope; (e) voltage-to-current converter; (f) voltage-stabilizing circuit; (g) sheet-bending loader.

**Figure 17 sensors-23-08393-f017:**
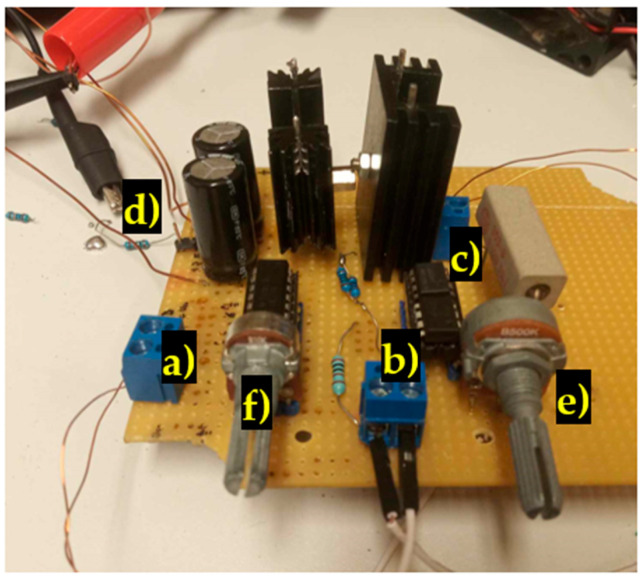
The designed voltage-controlled current source. (a) Terminal block for the secondary coil; (b) Terminal block for the control signal; (c) Terminal block for the primary coil; (d) Input power; (e) Current gain potentiometer; (f) Secondary voltage amplification gain potentiometer.

**Figure 18 sensors-23-08393-f018:**
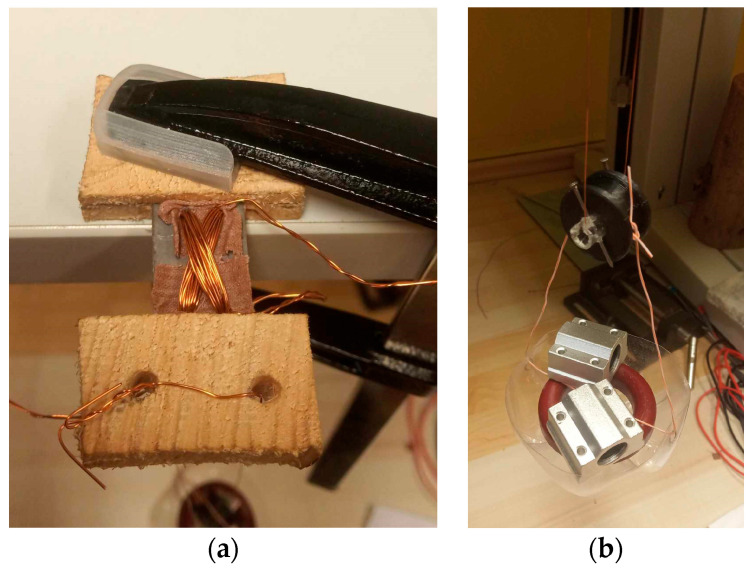
(**a**)—One side of the sensor was clamped to the table; and between the clamps and the sensor; wood was placed that is not ferromagnetic so that it will not affect the magnetic field inside the sensor. (**b**)—The sensor was loaded via a pulley that adjusted the load position if there were forces that would not act in the direction of gravity.

**Figure 19 sensors-23-08393-f019:**
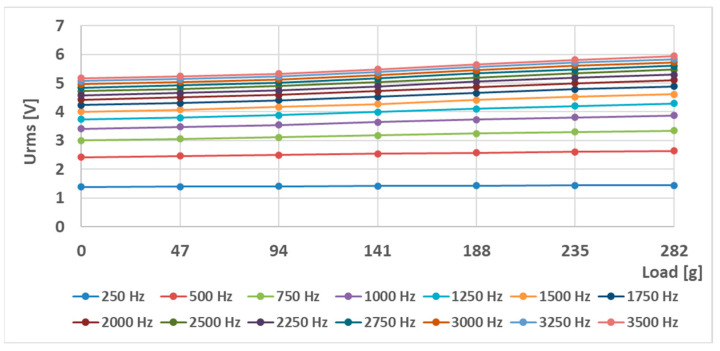
The output RMS voltage of the sensor (S-a) at various loadings at different sinusoidal magnetizing current frequencies at an amplitude of 180 mA.

**Figure 20 sensors-23-08393-f020:**
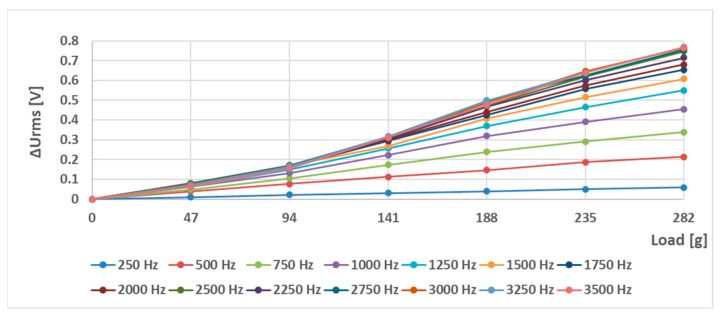
The relative changes in output RMS voltage of the sensor (S-a) at various loadings at different sinusoidal magnetizing current frequencies at an amplitude of 180 mA compared to the unstressed state.

**Figure 21 sensors-23-08393-f021:**
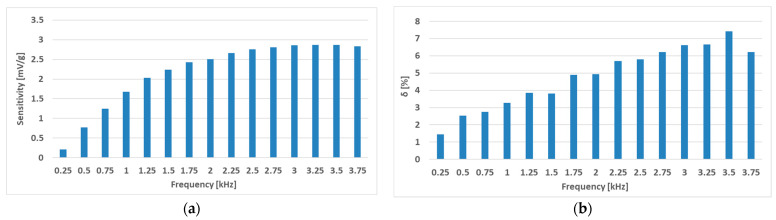
The sensitivity (**a**) and the linearity (**b**) of the characteristics as a function of the magnetizing current frequency of the (S-a) sensor. The largest sensitivity was observed at a frequency of 3.25 kHz (2.868 mV/g) and the highest linearity at a frequency of 250 Hz (1.45%).

**Figure 22 sensors-23-08393-f022:**
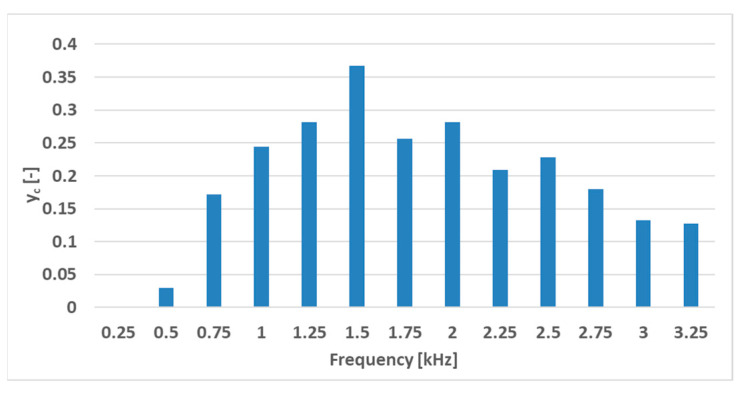
The combined characteristic of the sensor at various magnetizing current frequencies for the (S-a) sensor. The most favorable performance was observed at a frequency of 1.5 kHz with a *y_c_* value of approximately 0.357.

**Figure 23 sensors-23-08393-f023:**
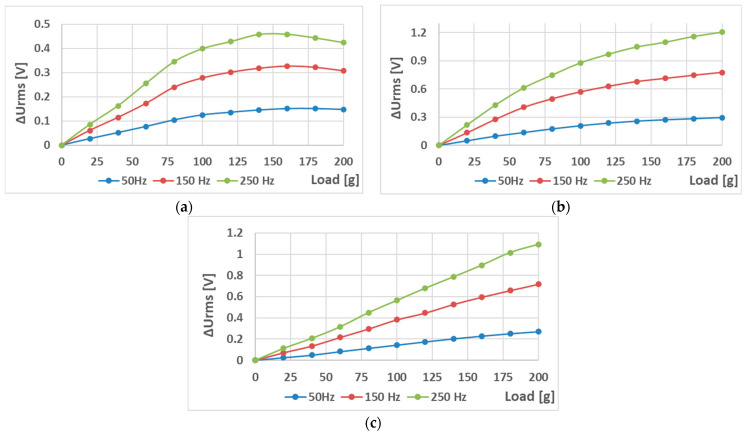
The static loading responses of the sensors have a magnetizing current amplitude of 300 mA. (**a**) sensor (S-a); (**b**) sensor (S-b); and (**c**) sensor (S-c). Sensor (S-b) is more linear than (S-a), and Sensor (S-c) is more linear than (S-b).

**Figure 24 sensors-23-08393-f024:**
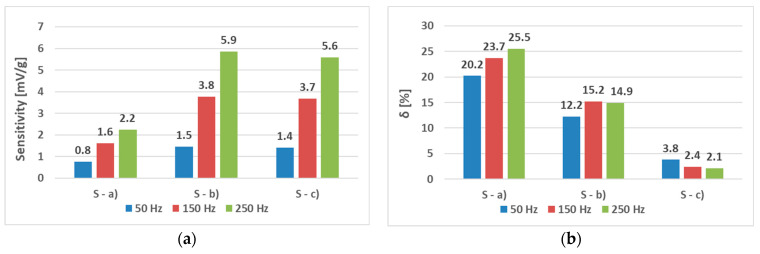
The sensitivity (**a**) and linearity (**b**) of the tested sensors at a current amplitude of 300 mA. The sensitivity rose with frequency in all three cases. The linearity error rose with sensor (S-a) as expected but was non-monotonous for sensor (S-b) and decreasing for sensor (S-c).

**Figure 25 sensors-23-08393-f025:**
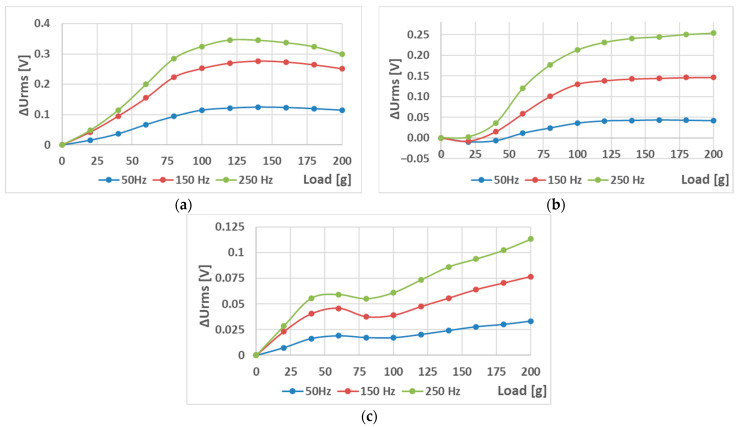
The static loading responses of the sensors at a magnetizing current amplitude of 100 mA. (**a**) sensor (S-a); (**b**) sensor (S-b); and (**c**) sensor (S-c). Compared to the 300 mA excitation graphs, they maintain the overall trend; however, a nonlinear region is present in all three cases. After the load reaches 100 g, the behavior is similar.

**Figure 26 sensors-23-08393-f026:**
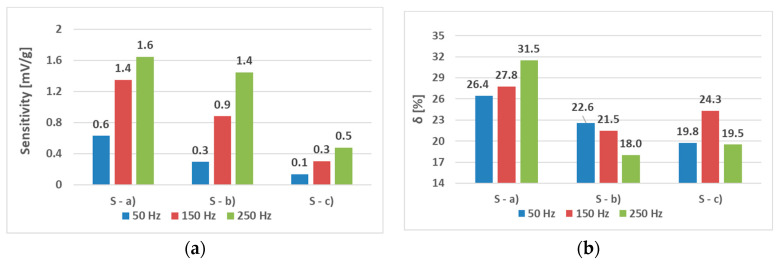
The sensitivity (**a**) and linearity (**b**) of the tested sensors at a current amplitude of 100 mA. The sensitivity still rose with frequency in all three cases; however, the linearity error showed a more complicated behavior, which was non-monotonous for sensor (S-c) and decreasing for sensor (S-b). The values were strongly influenced by the initial nonlinear region in Figure 25.

**Figure 27 sensors-23-08393-f027:**
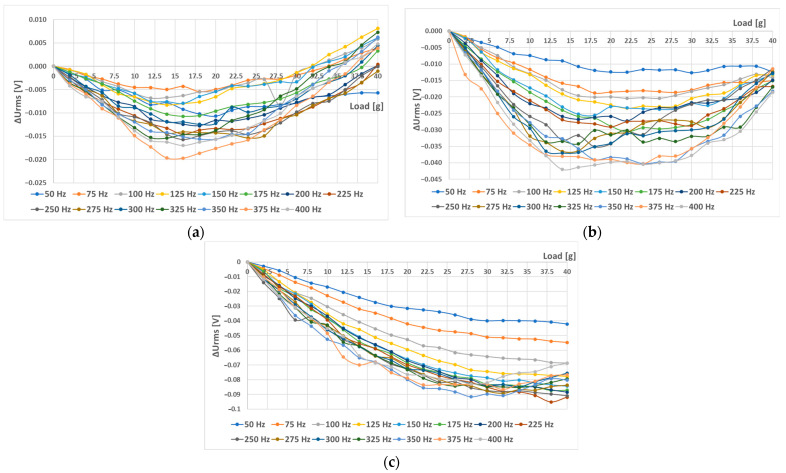
The static loading responses of the (L-a) sensor at different magnetizing current amplitudes and frequencies. (**a**) A total of 100 mA current amplitude; (**b**) 200 mA current amplitude; (**c**) 300 mA current amplitude.

**Figure 28 sensors-23-08393-f028:**
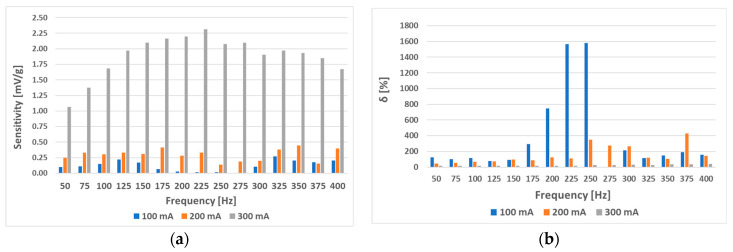
Sensitivity (**a**) and linearity (**b**) of the static loading characteristic of the (L-a) sensor. When the magnetizing current was 100 mA, the linearity error at some frequencies exceeded 1000%, which is evident from the strongly convex graph. As the magnetizing current amplitude increased, the linearity error decreased.

**Figure 29 sensors-23-08393-f029:**
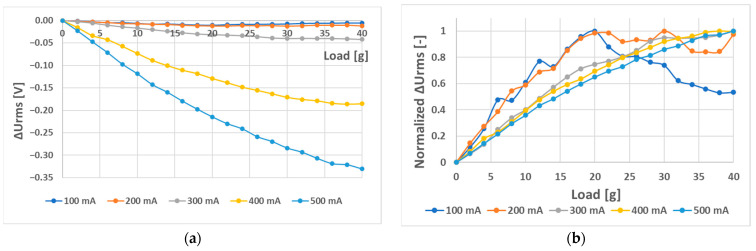
(**a**) Comparison of the static loading characteristic of sensor (L-a) at different current amplitudes. Graph (**b**) was created by normalizing graph (**a**) by dividing each data point by its largest *U_RMS_* change to show the linearity of the characteristic directly. A linear region in the interval from 0 to 14 g at all amplitudes was observed.

**Figure 30 sensors-23-08393-f030:**
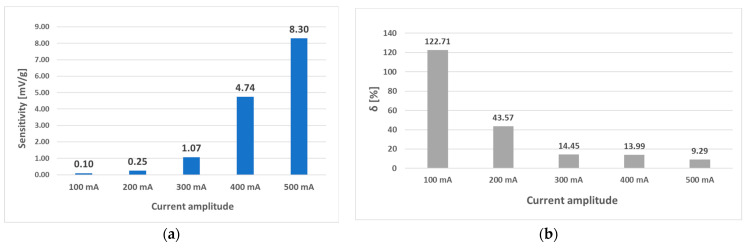
The sensitivity and linearity of the transfer characteristic at a frequency of 50 Hz of the (L-a) sensor, (**a**) sensitivity comparison, and (**b**) linearity error comparison. The strongly nonlinear behavior in the second half of the loading character caused the largest portion of the linearity error.

**Figure 31 sensors-23-08393-f031:**
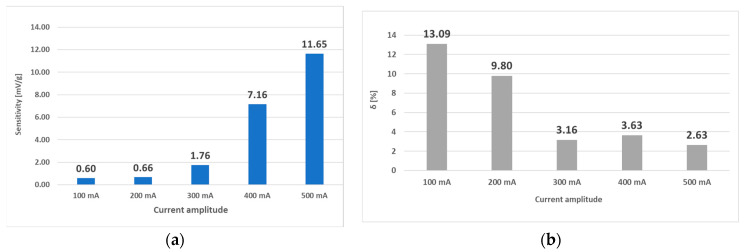
The sensitivity and linearity of the transfer characteristic at a frequency of 50 Hz of the (L-a) sensor only for the loading interval from 0 g to 14 g. (**a**)—sensitivity comparison; (**b**) linearity error comparison. The linearity error decreased, and sensitivity increased with increasing current amplitude. The linearity in the truncated region is more than four times higher.

**Figure 32 sensors-23-08393-f032:**
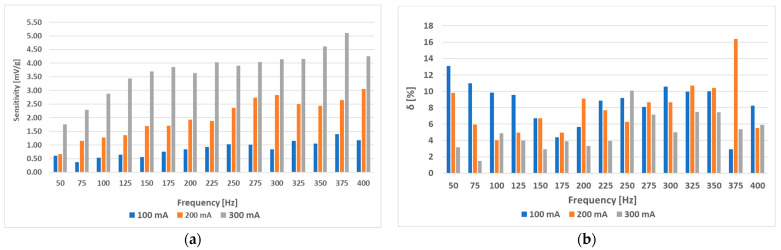
Sensitivity (**a**) and linearity (**b**) of the static loading characteristic of the (L-a) sensor within a load range of 0 g to 14 g. The linearity error in this region did not exceed 16.3%, and generally, increased current amplitudes made the linearity error smaller and the sensitivity larger.

**Figure 33 sensors-23-08393-f033:**
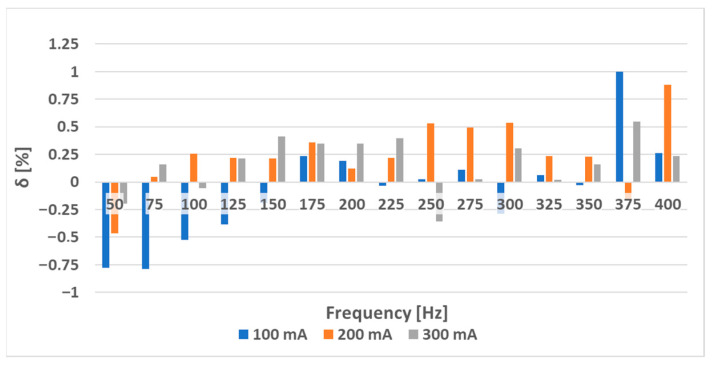
The combined characteristic for the static loading characteristic for loads between 0 g and 14 g when measuring the (L-a) sensor. The most favorable characteristics were obtained at the higher frequencies of 375 and 400 Hz.

**Figure 34 sensors-23-08393-f034:**
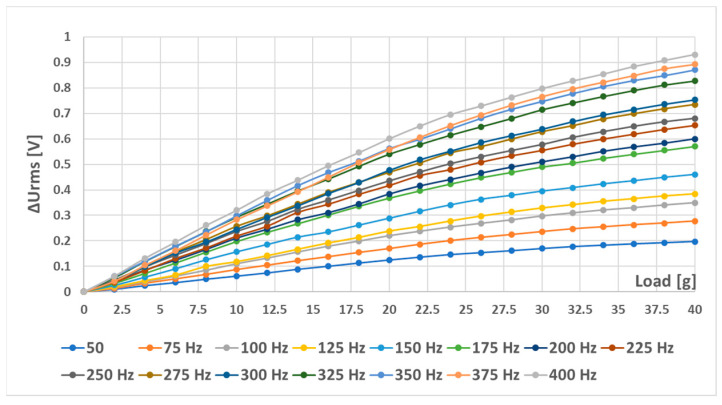
The static loading response of the (L-b) sensor at various frequencies and an amplitude of 300 mA.

**Figure 35 sensors-23-08393-f035:**
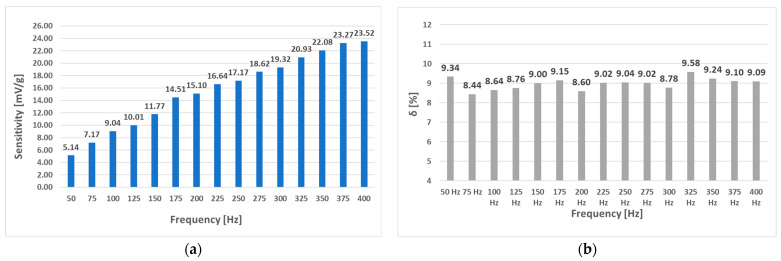
The sensitivity (**a**) and linearity (**b**) of the static loading characteristic of the (L-b) sensor at a current amplitude of 300 mA. Compared to the (L-a) sensor, the linearity error does not vary much with frequency and is kept at a nearly constant value of approx. 9%, therefore making the higher frequencies more favorable by creating a more sensitive response signal a) while not affecting the linearity of the loading characteristic.

**Figure 36 sensors-23-08393-f036:**
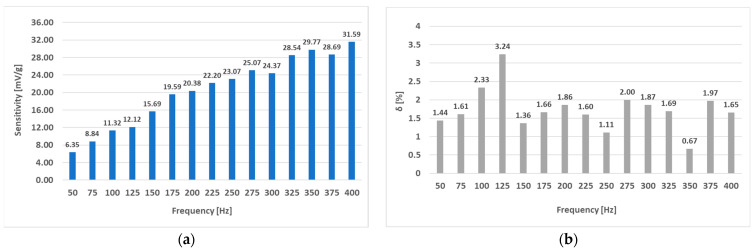
The sensitivity and linearity of the static loading characteristic of the (L-b) sensor at loadings from 0 g to 14 g and a current amplitude of 300 mA. Compared to the (L-a) sensor, the linearity error (**b**) does not vary much with frequency and is kept at approximately 2% on average. The sensitivity (**a**) rises approximately proportionally with frequency.

**Figure 37 sensors-23-08393-f037:**
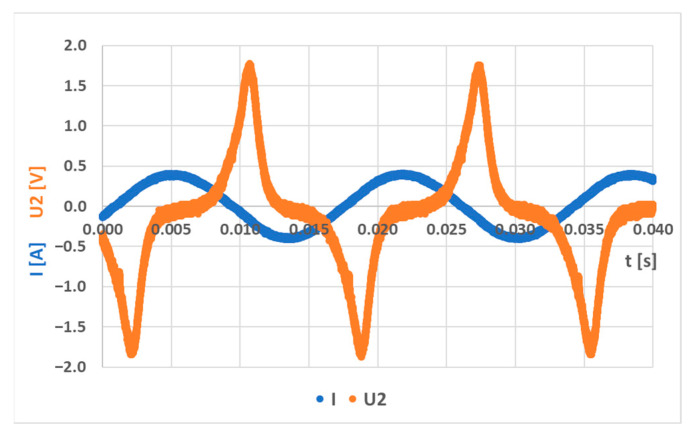
Current and voltage waveforms of the (L-a) sensor at a frequency of 60 Hz and a current amplitude of 415 mA and no load.

**Figure 38 sensors-23-08393-f038:**
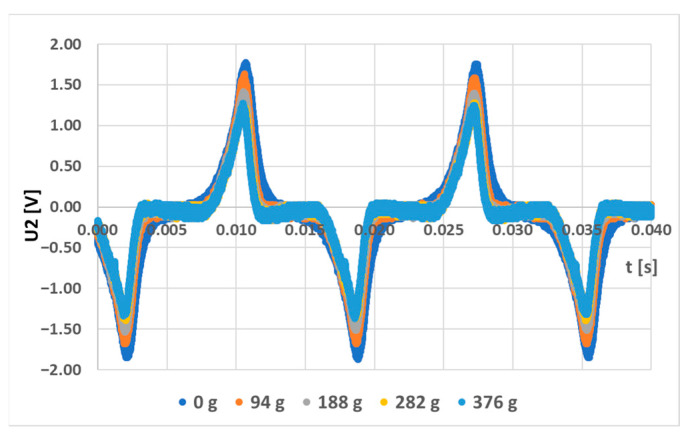
The output voltage changes on the secondary coil due to different loadings of the sensor.

**Figure 39 sensors-23-08393-f039:**
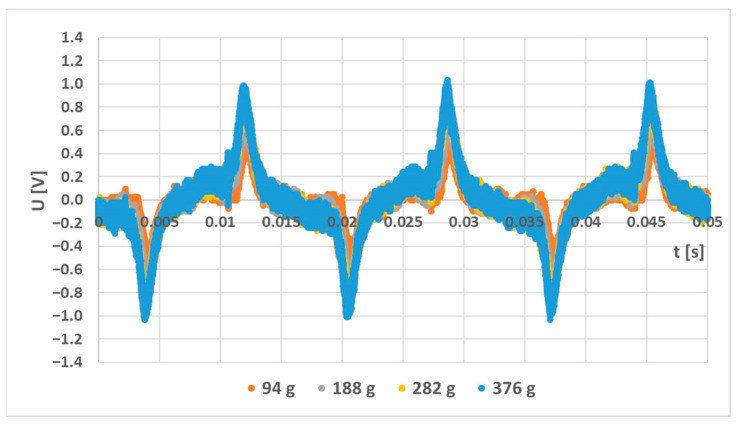
The voltage changes in the output voltage of the (L-a) sensor are due to increased mechanical loading.

**Figure 40 sensors-23-08393-f040:**
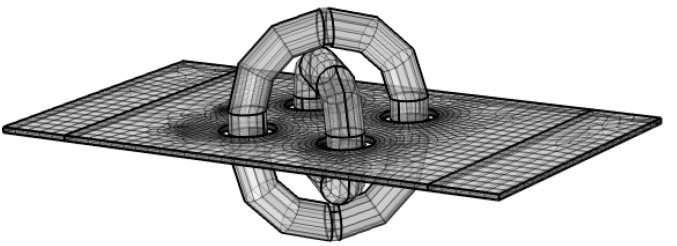
The generated mesh is the simulated sheet of the sensor. The surrounding air domain was hidden in this picture. The air formed a sphere with a radius two times larger than the sheet’s length.

**Figure 41 sensors-23-08393-f041:**
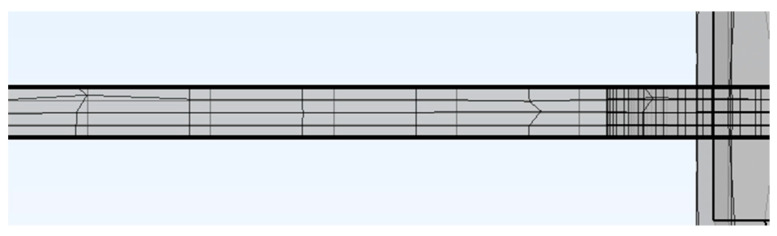
The detail of the mesh swept throughout the sheet’s thickness.

**Figure 42 sensors-23-08393-f042:**
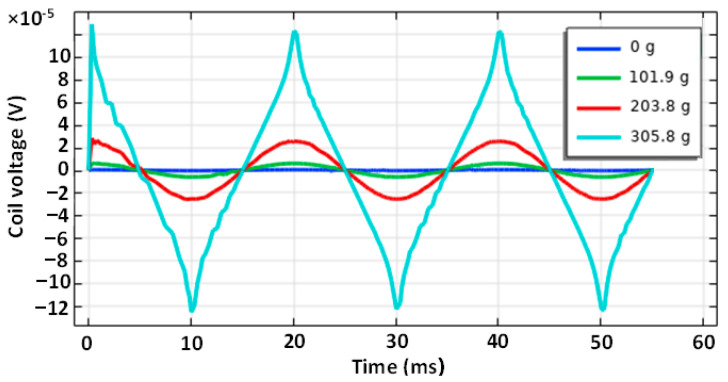
The induced voltage waveforms that were obtained by simulating the (L-a) sensor model that was measured experimentally yielded graph Figure 39. The induced voltage amplitude is smaller compared to the measured values, but at higher loadings, the peaks start to resemble the measured waveforms. The peaks in the measurement were a result of the inherent magnetic anisotropy, whereas in the simulation, the anisotropy was induced by the mechanical stress.

**Figure 43 sensors-23-08393-f043:**
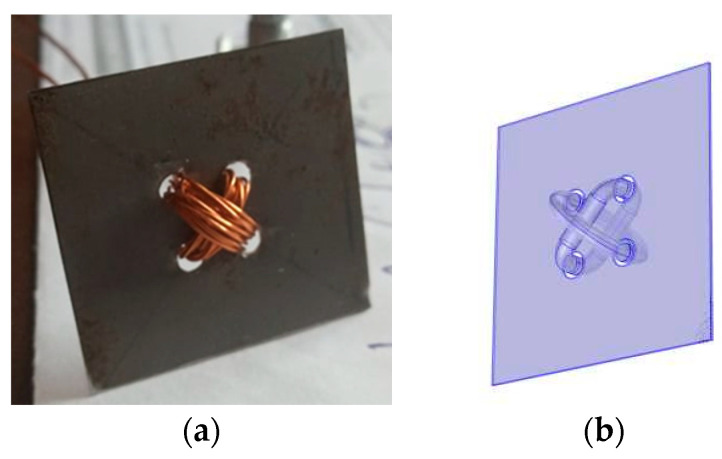
(**a**) Picture of the square sensor sample with a side length of 31 mm, and (**b**) 3D model of the sensor.

**Figure 44 sensors-23-08393-f044:**
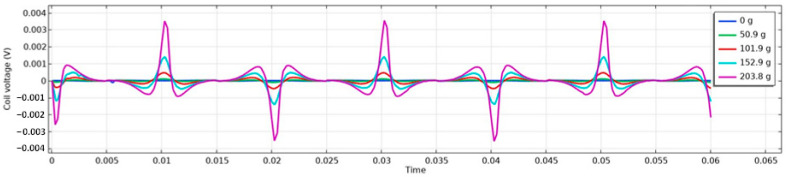
The secondary voltage of the simulated sensor at different loading forces.

**Figure 45 sensors-23-08393-f045:**
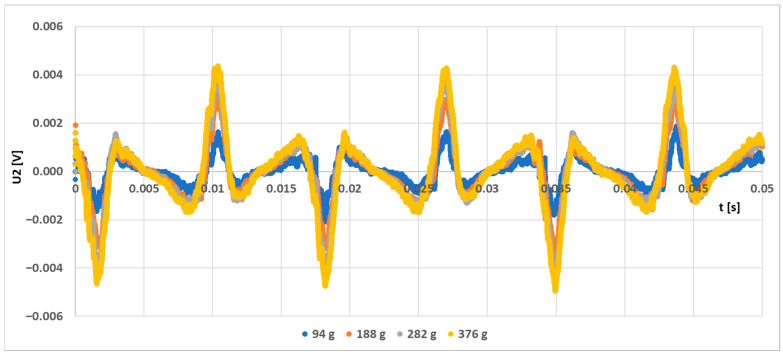
The secondary voltage of the measured real sensor at different loadings.

## Data Availability

The data presented in this study are available on request from the corresponding author.

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
