# Peer review of "Driving Signal and Geometry Analysis of a Magnetoelastic Bending Mode Pressductor Type Sensor"

_sensors, 2023, doi:10.3390/s23208393_

Round 1
Reviewer 1 Report
The main question addressed by the research of this manuscript is the analysis of a magnetoelastic bending mode sensor, specifically focusing on the effects of different factors such as the placement of holes in the windings, frequency of the magnetizing current waveform, and material characteristics on the sensor's linearity and sensitivity in measuring bending forces. The research aims to optimize the design and operation of these sensors for the measurement of small forces in the range of a few newtons.
In terms of originality and relevance, the research contributes to the field of magnetoelastic sensors by exploring the use of such sensors for measuring bending forces, which is less common compared to their typical application in measuring compressive forces. Additionally, the study investigates various design parameters and operational conditions that can enhance the linearity and sensitivity of the sensors. This can be considered relevant as it seeks to improve the performance of these sensors for specific applications.
The research also identifies some challenges and potential solutions related to the design and manufacturing of these sensors, such as the impact of drilling holes and cutting the sheet on sensor performance and the need for preventing sensor overload. Overall, the research addresses specific aspects of magnetoelastic sensor design and operation, potentially filling a gap in the field by providing insights into the optimization of such sensors for bending force measurement.
I think that the last section is traditionally titled as “Conclusions” instead of “Discussion”.
The photograph in Figure 16 shows an honest view of the typical laboratory desk. However, it would look nicer in a publication if it was more tidy!
The citations are relevant and numerous.
The editor can decide acceptance once these minor issues are resolved.
Author Response
Please see the attachment for the report.
Šimon Gans et al.

Reviewer 2 Report
Solid but large paper. Consider of the possibility moving some part of it in Supplementary Materials.
Some questions and advices:
Why Nelder-Mead optimization algorithm is chosen?
Is "%" is missed in (4)?
It is better to increase the font size in Fig. 2, 4, 6.
You can remove built-in titles from Fig. 20-42.
Some minor mistakes:
1. No need to copy affiliation, just type it once and write all e-mails.
2. Load [g] has strange placement in Fig. 20 and 21.
Author Response
Please see the attachment.
Šimon Gans et al.

Reviewer 3 Report
1- Try to clearly identify and highlight the research problem, discuss it in the light of the related work.
2- Then, the presented model needs to be discussed and evaluated in comparison to the state of the art.
3- English proofreading is necessary.
4- Please update the paper references, 2018 and above.
5- Future work must be included at the end of paper conclusion.
1. What is the main question addressed by the research?
This paper presents a brief overview of magnetoelastic sensors and defines the current state of the art. The main question addressed by the research is how to effectively measure bending forces using magnetoelastic sensors made from regular transformer sheets.
2. Do you consider the topic original or relevant in the field? Does it address a specific gap in the field?
To some extent, the use of magnetoelastic sensors is not entirely novel.
3. What does it add to the subject area compared with other published material?
The research problem not clear, no comparison to the state of the art.
4. What specific improvements should the authors consider regarding the methodology? What further controls should be considered?
- Please, try to clearly identify and highlight the research problem, discuss it in the light of the related work.
- Then, the presented model needs to be discussed and evaluated in comparison to the state of the art.
- English proofreading is necessary.
5. Are the conclusions consistent with the evidence and arguments presented and do they address the main question posed?
No, the paper conclusion is poorly written thus it would be beneficial to rephrase it in a way that effectively conveys the main findings of the paper. Future work must be included at the end of paper conclusion.
6. Are the references appropriate?
Yes, but the authors must update the references to be from 2018 to 2023.
7. Please include any additional comments on the tables and figures.
The figures in the paper need to be clearer and more legible.
1- Moderate editing of the English language is necessary.
Author Response

(The authors gave the same response as above.)

Round 2
Reviewer 3 Report
Accept in present form
English language fine